# CosR Regulation of *perR* Transcription for the Control of Oxidative Stress Defense in *Campylobacter jejuni*

**DOI:** 10.3390/microorganisms9061281

**Published:** 2021-06-11

**Authors:** Myungseo Park, Sunyoung Hwang, Sangryeol Ryu, Byeonghwa Jeon

**Affiliations:** 1Division of Environmental Health Sciences, School of Public Health, University of Minnesota, Minneapolis, MN 55455, USA; park2421@umn.edu; 2Department of Food and Animal Biotechnology, Research Institute for Agriculture and Life Sciences, Seoul National University, Seoul 08826, Korea; skyborn7@korea.kr; 3Department of Agricultural Biotechnology, Research Institute for Agriculture and Life Sciences, Seoul National University, Seoul 08826, Korea; 4Center for Food Bioconvergence, Seoul National University, Seoul 08826, Korea

**Keywords:** *Campylobacter*, oxidative stress, CosR, PerR

## Abstract

Oxidative stress resistance is an important mechanism to sustain the viability of oxygen-sensitive microaerophilic *Campylobacter jejuni*. In *C. jejuni*, gene expression associated with oxidative stress defense is modulated by PerR (peroxide response regulator) and CosR (*Campylobacter* oxidative stress regulator). Iron also plays an important role in the regulation of oxidative stress, as high iron concentrations reduce the transcription of *perR*. However, little is known about how iron affects the transcription of *cosR*. The level of *cosR* transcription was increased when the defined media MEMα (Minimum Essential Medium) was supplemented with ferrous (Fe^2+^) and ferric (Fe^3+^) iron and the Mueller–Hinton (MH) media was treated with an iron chelator, indicating that iron upregulates *cosR* transcription. However, other divalent cationic ions, such as Zn^2+^, Cu^2+^, Co^2+^, and Mn^2+^, did not affect *cosR* transcription, suggesting that *cosR* transcription is regulated specifically by iron. Interestingly, the level of *perR* transcription was increased when CosR was overexpressed. The positive regulation of *perR* transcription by CosR was observed both in the presence or in the absence of iron. The results of the electrophoretic mobility shift assay showed that CosR directly binds to the *perR* promoter. DNase I footprinting assays revealed that the CosR binding site in the *perR* promoter overlaps with the PerR box. In the study, we demonstrated that *cosR* transcription is increased in iron-rich conditions, and CosR positively regulates the transcription of PerR, another important regulator of oxidative stress defense in *C. jejuni*. These results provide new insight into how *C. jejuni* regulates oxidative stress defense by coordinating the transcription of *perR* and *cosR* in response to iron.

## 1. Introduction

*Campylobacter* spp. are a leading bacterial cause of gastroenteritis worldwide, accounting for approximately 166 million illnesses and 37,600 deaths per year [1]. Human infections with *Campylobacter jejuni*, the major pathogenic species of *Campylobacter*, may develop severe abdominal cramps, watery or bloody diarrhea [2], and, in some cases, induce Guillain–Barré syndrome, an acute flaccid paralysis [3]. Because *C. jejuni* is a commensal bacterium in the gastrointestinal tract of poultry, human campylobacteriosis is caused most frequently by the consumption of contaminated poultry [4]. *C. jejuni* requires low oxygen concentrations, such as 3–15%, for growth, but is sensitive to oxygen concentrations in normal aerobic conditions. Thus, aerotolerance plays an important role in the survival of *C. jejuni* during zoonotic transmission from poultry to humans [5,6,7], and oxidative stress defense is the key mechanism underlying the aerotolerance of *C. jejuni* [6,8].

Usually, genes encoding enzymes involved in the detoxication of reactive oxygen species (ROS) are present redundantly in the genomes of bacteria. However, *C. jejuni* possesses only a sole copy of genes encoding ROS detoxification enzymes, such as alkyl hydroperoxide reductase (AhpC), catalase (KatA), and superoxide dismutase (SodB). In *C. jejuni*, the expression of genes of oxidative stress defense is regulated mainly by PerR (peroxide response regulator) and CosR (*Campylobacter* oxidative stress regulator). OxyR and SoxRS are common regulators of oxidative stress defense in many Gram-negative bacteria [9,10]; however, their orthologs are not present in *C. jejuni*. In Gram-positive bacteria, PerR is the counterpart of the OxyR of Gram-negative bacteria, which regulates peroxide stress defense. *C. jejuni* is one of few Gram-negative bacteria that harbor PerR, not OxyR [11]. PerR is a repressor of genes encoding peroxide resistance enzymes (e.g., AhpC and KatA) by directly binding to their promoters [8,11]; thus, the inactivation of *perR* makes *C. jejuni* hyper-resistant to H_2_O_2_ by derepressing the transcription of *ahpC* and *katA*, also increasing aerotolerance in *C. jejuni* [8,11]. Despite the well-known function of PerR in peroxide stress defense, we previously demonstrated that *C. jejuni* PerR regulates the transcription of *sodB*, the sole gene involved in the detoxification of superoxide resistance [12], showing that PerR regulates both peroxide and superoxide stress defense in *C. jejuni.*

CosR is another important regulator of oxidative stress defense in *C. jejuni*. CosR is an OmpR-type response regulator, and its homologs are found predominantly in the bacteria of ε-*Proteobacteria*, such as *Campylobacter*, *Helicobacter,* and *Wolinella* [13]. In thermotolerant *Campylobacter* spp., such as *C. jejuni*, *Campylobacter coli*, and *Campylobacter lari*, CosR is an orphan response regulator because no potential sensor kinase is available in the vicinity of *cosR*; however, the sensor kinase CosS is present in non-thermotolerant *Campylobacter* spp., such as *Campylobacter fetus*, *Campylobacter concisus*, and *Campylobacter hominis* [14]. Because the function of CosR is essential for the viability of *C. jejuni*, a knockout mutation of *cosR* leads to cell death, and its knockout mutant cannot be constructed [13,15,16]. Using antisense-mediated gene knockdown, in our previous studies, we identified the regulon of CosR and characterized its function in the regulation of oxidative stress defense [13,17], discovering that CosR regulates the expression of a number of genes involved in oxidative stress defense, such as *ahpC*, *katA*, and *sodB* [13,17].

Several studies have reported that iron is a metal cofactor repressing *perR* in *C. jejuni* [11,18,19]. Increased iron levels derepress PerR-regulated genes of oxidative stress defense, enabling *C. jejuni* to respond to oxidative stress. Despite the important roles played by CosR in the regulation of oxidative stress defense in *C. jejuni*, little has been studied about the association of CosR with iron and PerR to modulate the regulation of oxidative stress defense in *C. jejuni*. Aiming to fill this knowledge gap, in this study, we investigated how iron affects *cosR* transcription and how CosR is related to *perR* transcription.

## 2. Materials and Methods

### 2.1. Bacterial Strains and Culture Conditions

*C. jejuni* NCTC 11168 and its derivatives were grown at 42 °C in Mueller–Hinton (MH) media (Difco) or MEMα (Minimum Essential Medium; Gibco, Catalog no. 41061) under microaerobic conditions (5% O_2_, 10% CO_2_, and 85% N_2_). MEMα is a defined medium that is commonly used to control iron concentrations in culture media for *C. jejuni* [20,21,22]. Iron levels were controlled by growing *C. jejuni* in MEMα supplemented with iron or by treating MH media with deferoxamine mesylate (DFMS), an iron chelator [23,24]. For the broth culture, an overnight culture on MH agar was resuspended in 3 mL of MH broth or MEMα to an OD_600_ of 0.08, and the bacterial suspension was microaerobically grown with shaking at 200 rpm. Kanamycin (50 μg mL^−1^) was occasionally added to the culture media to maintain pMW10 [25] and the P*_cosR_*::*lacZ* and P*_perR_*::*lacZ* promoter fusion constructs.

### 2.2. CosR Knockdown and Overexpression

Since *cosR* is essential for the viability of *C. jejuni* [13,15,16], instead of using gene knockout, the intracellular level of CosR was reduced using gene knockdown with antisense peptide nucleic acids (PNA), as described in our previous studies [13,17]. Briefly, CosR-specific PNA (CATTTGTTCTATCCTT), which binds reverse complementarily to the leader sequence spanning the ribosomal binding site and the start codon of *cosR* [13,17], was commercially synthesized by PNA Bio (Thousand Oaks, CA, USA). To improve cell permeability, the PNA was conjugated to the permeabilization oligonucleotide (KFFKFFKFFK), as reported previously [13,17]. Overnight cultures of *C. jejuni* grown on MH agar plates were resuspended in culture media to an optical density at 600 nm (OD_600_) of 0.07, and CosR-specific PNA was added to the suspension to a final concentration of 1.5 µM at the beginning of culture. The intracellular level of CosR was increased using a CosR-overexpression strain that was constructed in our previous study by integrating an extra copy of *cosR* into the chromosome of *C. jejuni* [17].

### 2.3. Construction of a P_cosR_::lacZ Promoter Fusion and β-Galactosidase Assay

The *cosR-lacZ* promoter fusion was constructed using pMW10, a promoterless *lacZ* shuttle vector [25]. The *cosR* promoter and its partial coding region were amplified with the primer pairs of CosR_PF_F(*Xba*I): CCCTTGAAGAGTCTAGAGACTTTGTAAGCTT and CosR_PF_R(*Xba*Ⅰ): CAAGCATCTAGACATACGCAGTCTTTTGTAA). The PCR product was cloned into pMW10 after digestion with *Xba*I, and the final construct was confirmed with sequencing. The constructed plasmid was introduced to *C. jejuni* NCTC 11168 by conjugation [25]. The *perR-lacZ* promoter fusion was constructed in our previous study [18]. β-Galactosidase assays were performed, as described previously, with some modifications [18,25,26]. Briefly, 80 µL of bacterial culture and 120 µL of the β-galactosidase assay mix, consisting of 60 mM Na_2_HPO_4_, 40 mM NaH_2_PO_4_, 10 mM KCl, 1 mM MgSO_4_, 36 mM β-mercaptoethanol, 1.1 mg/mL ONPG, and 6.7% PopCulture reagent (MilliporeSigma, St. Louis, MI, USA) were mixed and transferred into a 96-well plate to measure OD_420_ and OD_600_. After reading the OD_600_, the plate was incubated at 35 ℃ with shaking, and OD_420_ was measured every 10 min for 1 h in a plate reader (Varioskan, ThermoFisher, Waltham, MA, USA). Occasionally, the defined culture medium MEMα was supplemented with different concentrations of FeSO_4_, CoCl_2_, CuCl_2_, MnCl_2_, and ZnCl_2_, which were purchased from MilliporeSigma (St. Louis), to examine the effects of metal ions on the transcription of *cosR*.

### 2.4. Electrophoretic Mobility Shift Assay

An electrophoretic mobility shift assay (EMSA) using recombinant CosR (rCosR) was performed, as described previously [13,17]. Briefly, *Escherichia coli* BL21 (DE3) carrying plasmid pET15b::*cosR* was grown to an OD_600_ of approximately 1.0. After induction with 0.5 mM IPTG for 3 h, rCosR was purified under native conditions using Ni^2+^ affinity chromatography. The DNA fragments containing the promoter region of *perR* were PCR-amplified with the primer pairs of *perR*_F: AGACAAATTTATTGAACATGGAAAAACAAG and *perR*_R: AGAGATTGAAGGGTATTCTTTTTTAATTTC, purified from agarose gel using a gel extraction kit (Qiagen, Hilden, Germany), and labeled with [γ-^32^P] ATP (GE Healthcare, Chicago, IL, USA). After elimination of the unincorporated radioisotope with a MicroSpinTMG-25 column (GE Healthcare), the 0.2 nM of ^32^P-labeled DNA probe was incubated with the purified rCosR protein at different concentrations (0, 0.8, 1.6, 2.4, and 3.2 nM) at 37 °C for 15 min in 10 μL of the gel-shift assay buffer (20 mM HEPES (pH7.6), 1 mM EDTA, 10 mM (NH_4_)_2_SO_4_, 5 mM DTT, 0.2% Tween 20, 30 mM KCl, 0.1 μg poly (dI-dC)). Unlabeled PCR amplicon of the *perR* promoter, which was prepared with the same method as above without [γ-^32^P] ATP, was used as a competitor. The reaction mixtures were resolved in a 6% polyacrylamide gel, and the radiolabeled DNA fragments were visualized using the BAS2500 system (Fuji Film, Kyoto, Japan).

### 2.5. DNase I Footprinting Assay

A DNase I footprinting assay was performed following a method described previously [13,18]. DNA fragments containing the *perR* promoter region were PCR amplified using a ^32^P-labeled primer *perR*_FP_F: AGCCTTGCAAGAAATGAATAATAATGC and an unlabeled primer *perR*_FP_R: ATTCATCAATATTAGGATGCTCATGTC, and were purified from the agarose gel with Wizard SV Gel and the PCR Clean-Up System (Promega). Binding of rCosR to the ^32^P-labeled *perR* promoter was performed at 37 °C for 10 min in 40 μL of the gel-shift assay buffer (20 mM HEPES (pH7.6), 1 mM EDTA, 10 mM (NH_4_)_2_SO_4_, 5 mM DTT, 0.2% Tween 20, 30 mM KCl, 0.1 μg poly (dI-dC)) containing 10 mM of MgCl_2_. After treatment of the reaction mixture with or without 0.1 U DNase I (Takara), the reactions were stopped by the addition of 200 μL of ice-cold stop solution (0.4 M NaOAc, 2.5 mM EDTA), and the DNA products were purified by phenol extraction and ethanol precipitation. The digested DNA fragments were separated by electrophoresis in 6% polyacrylamide-8 M urea gels alongside sequencing ladders that were generated with the same ^32^P-labeled primer used to amplify DNA fragments for DNase I digestion.

## 3. Results

### 3.1. Regulation of cosR Transcription by Iron

The level of *cosR* transcription was measured with β-galactosidase assays by supplementing the defined medium MEMα with different concentrations of iron. To control iron concentrations in media, we used MEMα, a defined media that have been frequently used to analyze the effects of iron and other metal ions, such as Zn, on the physiology and pathogenicity of *C. jejuni* in many studies [20,21,22,27] The results of the P*_cosR_*-*lacZ* fusion assay demonstrated that iron increased the level of *cosR* transcription in a concentration-dependent manner (Figure 1). The level of *cosR* transcription was increased by 27.5% and 28% at 2 μM Fe^3+^ and 2 μM Fe^2^^+^, respectively, compared to the control without iron (Figure 1), suggesting that both Fe^3+^ and Fe^2+^ affect *cosR* transcription. The assay was also conducted with different divalent cationic ions, such as Zn^2+^, Cu^2+^, Co^2+^, and Mn^2+^, to examine whether *cosR* transcription was influenced by other metal ions. However, the transcriptional level of *cosR* was not changed by the divalent cationic ions (Figure 2), suggesting that *cosR* transcription is regulated specifically by iron.

### 3.2. Iron Regulation of cosR and perR Transcription over the Growth of C. jejuni

The levels of *perR* and *cosR* transcription were measured in the presence and absence of iron over the growth of *C. jejuni*. Iron levels were controlled by supplementing MEMα with Fe^3+^ or treating the MH media with an iron chelator. Regardless of iron, the transcriptional levels of both *perR* and *cosR* were reduced in the lag phase for the first few hours of culture MEMα, and increased in the exponential phase (Figure 3a,b). Because the effect of iron on *perR* transcription has been reported in several studies [11,18,19], P*_perR_*::*lacZ* was included as a control to compare the expression patterns with *cosR*. The increase in the level of *cosR* transcription by iron was significant both in the exponential and in the stationary phases in MEMα (Figure 3a), and similar patterns were observed when the experiment was performed in MH media using an iron chelator (Figure 3c). After cultivation in iron-rich conditions for 8 h, the levels of *cosR* transcription were increased by 21.1% and 40.4% in MEMα and MH media, respectively (Figure 3a,c), whereas *perR* transcription was reduced by 22.2% and 47.9% in MEMα and MH media, respectively (Figure 3b,d).

### 3.3. CosR Regulation of perR Transcription

Based on the role of CosR and PerR in the regulation of oxidative stress defense, we hypothesized that the two regulators may influence their transcription to coordinate the transcriptional regulation of oxidative stress defense. To examine this hypothesis, we measured whether CosR might affect *perR* transcription. Due to the essentiality of CosR in the viability of *C. jejuni*, a knockout mutant of *cosR* could not be constructed in multiple studies [13,15,16]. Thus, we measured the level of *perR* transcription under the conditions of CosR overexpression and CosR knockdown which were established in our previous studies [13,17]. Interestingly, the results of the P*_perR_*-*lacZ* fusion assay showed that the level of *perR* transcription was increased when CosR was overexpressed (Figure 4). However, CosR knockdown by antisense PNA did not affect *perR* transcription significantly (Figure 4). Although the level of *perR* transcription was increased by CosR overexpression in the presence and absence of iron, the overall level of *perR* transcription was reduced by iron regardless of the levels of CosR in wild-type, a CosR-overexpression strain, and CosR-knockdown conditions (Figure 4), suggesting that CosR regulation of *perR* transcription is independent of iron.

### 3.4. CosR Binding to the perR Promoter

Since CosR positively affected *perR* transcription (Figure 4), we examined whether CosR regulation of *perR* transcription is mediated by direct interaction between CosR and the *perR* promoter. The binding of CosR to the *perR* promoter was examined with EMSA, which showed that CosR was directly bound to the *perR* promoter (Figure 5a). Previously, we reported that the transcription of *perR* is driven by two promoters, and PerR regulates its transcription by autoregulation [18]. By performing a DNase I footprinting assay, we identified the CosR binding site in the −10 and −35 region of the two *perR* promoters (Figure 5b), and a part of the 5′ region of the CosR binding site that overlapped with the PerR box to which PerR binds for autoregulation (Figure 5c). These results indicate that CosR regulates *perR* transcription by directly binding to the *perR* promoter.

## 4. Discussion

Since *C. jejuni* is a microaerophilic bacterium sensitive to oxygen levels in normal atmospheric conditions, oxidative stress defense is important for the survival of *C. jejuni* during foodborne transmission to humans through various routes that generally involve aerobic environments. To achieve a timely response to oxidative stress, oxidative stress defense should be coordinated efficiently in which CosR, PerR, and iron play key roles. Iron is an essential micronutrient affecting bacterial growth [28], and the unavailability of iron significantly reduces the growth of *C. jejuni* [22,24]. Moreover, iron participates in the generation of ROS through the Fenton reaction and is also involved in the regulation of oxidative stress defense [22,24], primarily by repressing the transcription of *perR* in *C. jejuni* [11,18]. PerR is a metal-responsive repressor protein [11,29]. Whereas *Bacillus subtilis* PerR uses either manganese or iron as a regulatory metal cofactor to detect oxidative stress [30,31], the transcription of *perR* in *C. jejuni* is affected only by iron, not manganese [18]. Similarly, the level of *cosR* transcription was increased by iron (Figure 1); however, other divalent cationic ions, such as Zn^2+^, Cu^2+^, Co^2+^, and Mn^2+^, did not affect *cosR* transcription (Figure 2).

Interestingly, iron affects the transcription of *perR* and *cosR* in the opposite pattern; iron-rich conditions reduce *perR* transcription but increase *cosR* transcription. The opposite patterns in the transcriptional regulation of *perR* and *cosR* by iron may facilitate the timely response of *C. jejuni* to oxidative stress defense because PerR is a repressor of ROS detoxification enzymes, such as AhpC, KatA, and SodB, whereas CosR is a positive regulator of AhpC and KatA [13,17]. Increased iron levels, which can be accompanied by an increase in oxidative stress, reduce the level of *perR* transcription, resulting in the depression of *ahpC*, *katA*, and *sodB*. (Figure 6). Under iron-rich conditions, however, *cosR* transcription was increased (Figure 1 and Figure 2), which may consequently increase the levels of AhpC and KatA due to the positive regulation by CosR (Figure 6). In response to increased iron levels, collectively, *C. jejuni* can stimulate the transcription of the genes encoding ROS detoxification enzymes through the negative regulation of *perR* transcription and the positive regulation of *cosR* transcription, leading to the alleviation of oxidative stress (Figure 6).

In this study, we discovered that CosR positively regulates *perR* transcription by directly binding to the *perR* promoter (Figure 4 and Figure 5). CosR regulated *perR* transcription only when the level of CosR was increased because CosR knockdown using antisense PNA did not affect *perR* transcription (Figure 4). Although *perR* transcription was positively regulated by CosR without iron, the level of *perR* transcription was increased by CosR overexpression more significantly in the presence of iron than the absence of iron (46.5% increase with iron vs. 14.5% increase without iron) (Figure 4), which can be ascribed to the positive regulation of *cosR* transcription by iron. High iron concentrations enhance the expression of ROS detoxification enzymes by repressing *perR* transcription and increasing *cosR* transcription. However, a regulatory mechanism is needed to coordinate the transcriptional levels of *perR* and *cosR* to avoid potential over-stimulation of ROS detoxification systems where CosR regulation of *perR* transcription may play a role.

The transcription of *perR* is driven by two adjacently located promoters, both of which are repressed by iron and subjected to autoregulation [18]. The CosR binding region that was identified using a DNase I footprinting assay overlapped with the PerR box to which PerR binds for autoregulation (Figure 5c). This suggests that PerR and CosR may compete for binding to the *perR* promoter under certain circumstances.

In the gastrointestinal tract, relatively high concentrations (ca. 25 mM) of iron are present theoretically; however, only a small proportion of iron is bioavailable because of the low water solubility of inorganic iron [32]. The acquisition of iron affects *C. jejuni* colonization of the gastrointestinal tracts since the knockout mutation of genes involved in iron acquisition results in colonization defects [33,34,35]. Because oxidative stress defense is a complicated process involving a number of factors, such as the presence of oxygen, iron, and antioxidants, it is difficult to speculate how the findings in this study can impact the regulation of oxidative stress defense, particularly during the colonization of the gastrointestinal tract, where oxygen levels are extremely low. Presumably, the regulation of oxidative stress defense involving CosR, PerR, and iron may contribute to the survival of this microaerophile when exposed to aerobic environments. In our previous study, we demonstrated that iron stimulates biofilm formation in *C. jejuni* by increasing oxidative stress, suggesting that iron utilization in combination with oxidative stress contributes to the survival of *C. jejuni* in aerobic environments [20]. At this stage, future studies are still needed to elucidate how oxidative stress defense is regulated to improve the survival of this fastidious bacterium under stress conditions during transmission and infection.

## Figures and Tables

**Figure 1 microorganisms-09-01281-f001:**
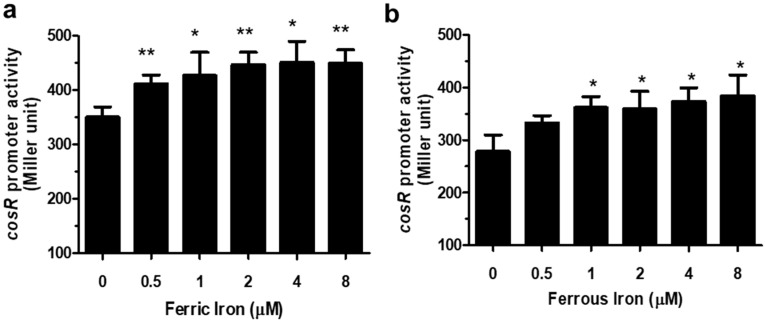
Positive regulation of *cosR* transcription by iron. The P*_cosR_*::*lacZ* fusion assay was conducted by supplementing MEMα with different concentrations of Fe^3+^ (**a**) and Fe^2+^ (**b**). The results show the means and standard deviations of three samples in a single experiment. The experiment was repeated three times, and similar results were obtained in the repeated experiments. The statistical analysis was performed using Student’s t-test in comparison with the control without iron. *: *p* < 0.05, **: *p* < 0.01.

**Figure 2 microorganisms-09-01281-f002:**
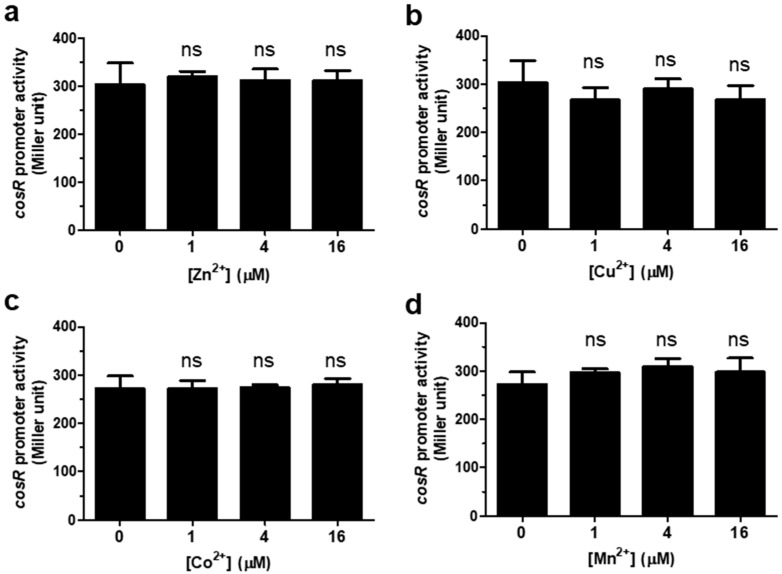
Irresponsiveness of *cosR* transcription to divalent metal ions other than iron. The P*_cosR_*::*lacZ* fusion assay was conducted with Zn^2+^ (**a**), Cu^2+^ (**b**), Co^2+^ (**c**), and Mn^2+^ (**d**). The results show the means and standard deviations of three samples in a single experiment. The experiment was repeated three times, and similar results were obtained in the repeated experiments. The statistical analysis was performed using Student’s *t*-test in comparison with the control without ions. ns: not significant.

**Figure 3 microorganisms-09-01281-f003:**
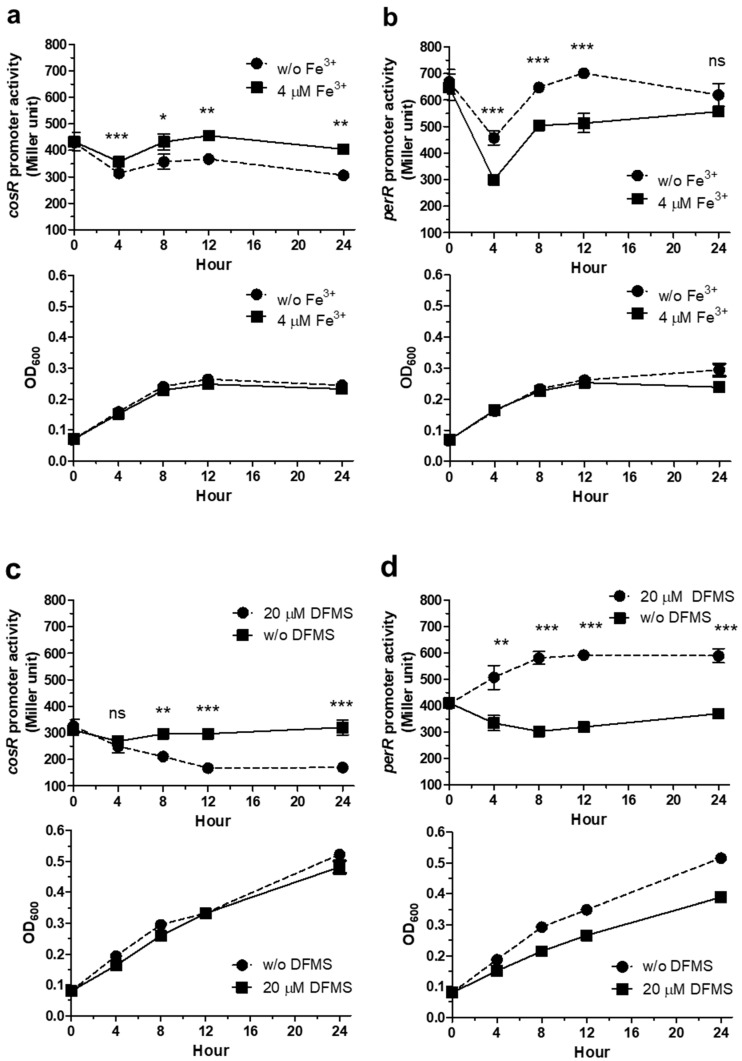
Effects of iron on the transcription of *cosR* and *perR* over the growth of *C. jejuni*. The P*_cosR_*::*lacZ* and P*_perR_*::*lacZ* fusion assays were conducted in the presence and absence of iron in MEMα and MH media: (**a**) P*_cosR_*::*lacZ* assay in MEMα. (**b**) P*_perR_*::*lacZ* in MEMα. (**c**) P*_cosR_*::*lacZ* assay in MH. (**d**) P*_perR_*::*lacZ* in MH. Deferoxamine mesylate (DFMS) was used as an iron chelator in MH media. The growth of *C. jejuni* is indicated with the optical density at 600 nm (OD_600_) in each panel. The experiment was repeated three times, and similar patterns of results were obtained in the experiments. The statistical analysis was performed using Student’s *t*-test by comparing the samples with and without iron at the same sampling point. *: *p* < 0.05, **: *p* < 0.01, ***: *p* < 0.001.

**Figure 4 microorganisms-09-01281-f004:**
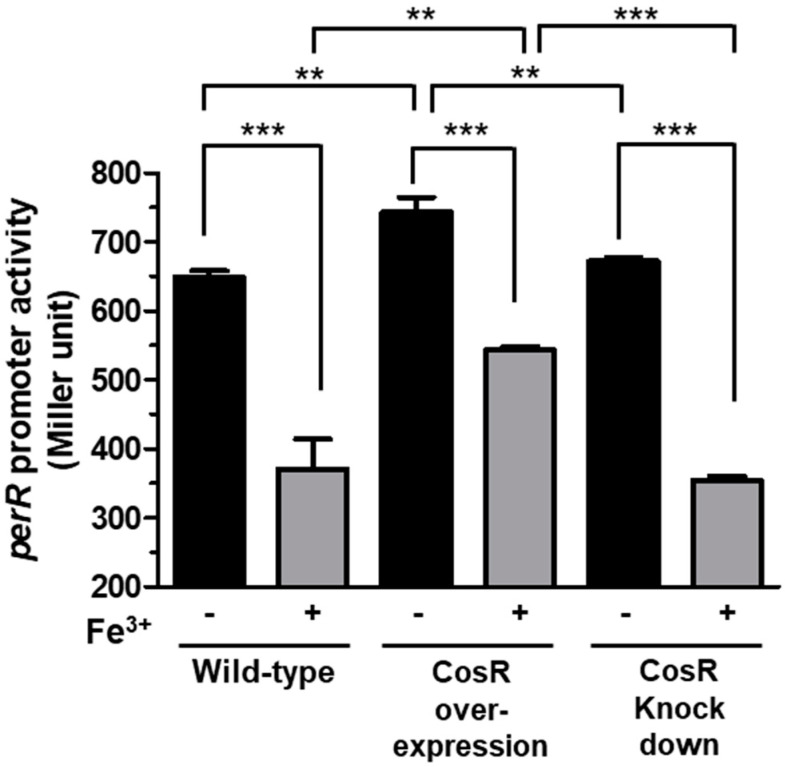
Positive regulation of *perR* transcription by CosR. The P*_perR_*::*lacZ* fusion assays were conducted under conditions with different levels of CosR, including wild-type, CosR overexpression, and CosR knockdown using antisense PNA. The experiment was repeated three times, and similar results were obtained in the experiments. The statistical analysis was performed using Student’s *t*-test. **: *p* < 0.01, ***: *p* < 0.001.

**Figure 5 microorganisms-09-01281-f005:**
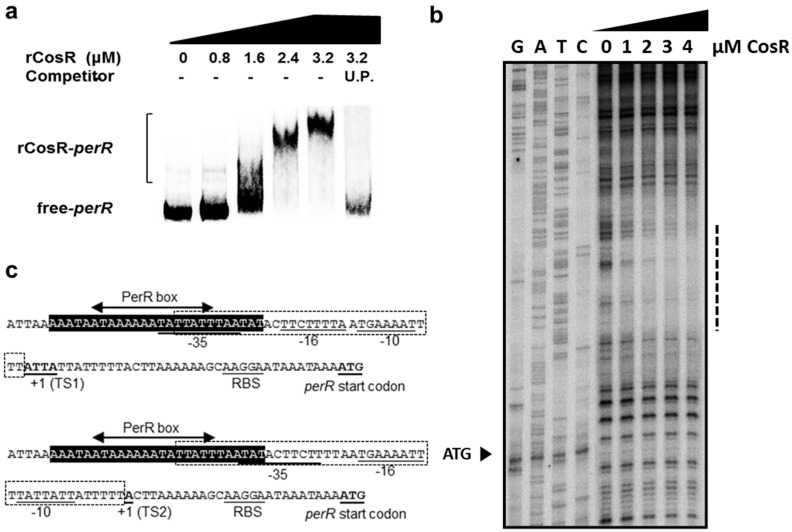
CosR binding to the *perR* promoter (**a**) The results of EMSA showed that rCosR bound to the *perR* promoter. Unlabeled probe (U.P.) is the PCR amplicon of the *perR* promoter, which was prepared without [γ-32P] ATP and was used as a competitor. (**b**) Identification of the CosR binding sites in the *perR* promoter by DNase I footprinting. The CosR binding region is indicated with a dotted line, and ATC is the start codon of *perR*. (**c**) The CosR binding site in the two adjacently located *perR* promoters, which were reported in a previous study [18]. The PerR box is the site for PerR binding for autoregulation [18]. The *perR* start codon, the transcriptional start site (+1), and the −10, −16, and −35 elements are underlined.

**Figure 6 microorganisms-09-01281-f006:**
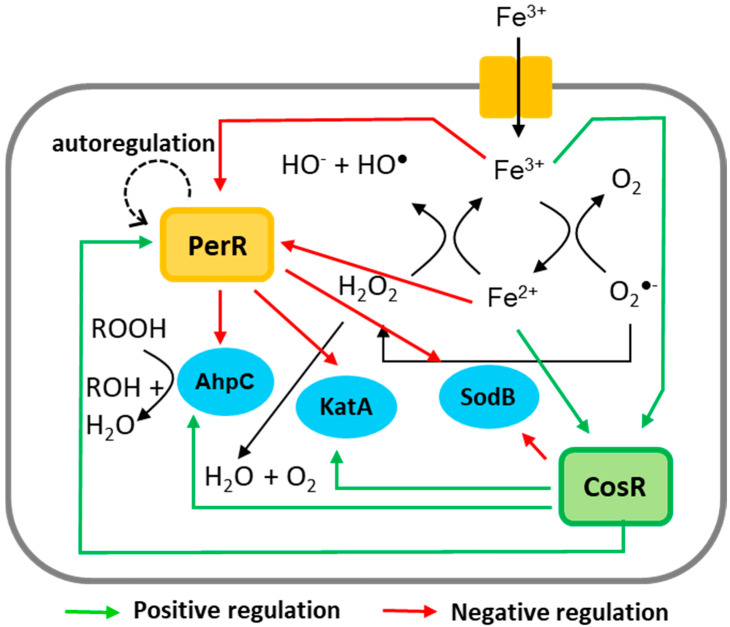
Schematic diagram of oxidative stress defense regulation by CosR and PerR. Positive and negative regulations are indicated with green and red lines, respectively.

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
