# Peer review of "CosR Regulation of perR Transcription for the Control of Oxidative Stress Defense in Campylobacter jejuni"

_microorganisms, 2021, doi:10.3390/microorganisms9061281_

Round 1

Reviewer 1 Report

The manuscript by Park et al. delivers information about the interplay of the two main regulators CosR and PerR involved in oxidative stress defense in C. jejuni. Using EMSA and footprint analysis as well as lacZ-fusion assays, the authors observed that CosR slightly stimulated perR expression in an iron-independent manner by direct binding to the perR promoter region. cosR expression also seemed to be somewhat influenced by iron concentration.

Major points:

  1. Fig.1: Since C. jejuni only marginally grows in MEMalpha (Fig. 3, only 1-2 cell divisions), it would be important to add data on cosR promotor activity under growth conditions in complex medium relative to the data in MEMalpha (as additional “high iron concentration” dataset).
  2. Do the authors know the intrinsic concentration of iron in MEMalpha? Does C. jejuni grow without any iron?
  3. Fig. 3: Experimental data should be added for growth in complex medium, in order to evaluate the effect of growth phase, since C. jejuni only doubles once or twice over 12-24 hours in MEMalpha (see point 1.), conclusions about growth phase are not easy to draw.
  4. Line 159 (“data not shown”): data for Fe2+ should be added as a second group of bars in Fig. 1
  5. The authors should discuss their findings with respect to the concentration of iron in the intestinal environment.

Minor points:

  1. Line 103: The construction of the cosR overexpression strain was described in reference [13] rather than [17].
  2. Lines 249-250: The authors should add some notion about the magnitude of iron affecting cosR and perR expression.

Author Response

Reviewer 1

Major points:

  • Fig.1: Since C. jejuni only marginally grows in MEMalpha (Fig. 3, only 1-2 cell divisions), it would be important to add data on cosR promotor activity under growth conditions in complex medium relative to the data in MEMalpha (as additional “high iron concentration” dataset).

Response: MEMα is defined media that are commonly used to examine the effects of iron and other metal irons on Campylobacter [1-4]. C. jejuni grows better in rich media, such as MH and BHI; however, MEMα provides C. jejuni with amino acids, the major carbon source for the growth of C. jejuni. Considering the reviewer’s comment, we included data that was generated using MH media in the revised version (Figure 3).

  • Do the authors know the intrinsic concentration of iron in MEMalpha? Does C. jejuni grow without any iron?

Response: MEMα does not contain iron. That is why the media are used to examine the effects of iron and some other trace metal ions on the physiology and pathogenicity of Campylobacter in many studies [1-4]. It is thought that low levels of intracellular iron within Campylobacter from inoculums help sustain bacterial growth in MEMα, although the levels are lower than iron levels in rich media.   

  • Fig. 3: Experimental data should be added for growth in complex medium, in order to evaluate the effect of growth phase, since C. jejuni only doubles once or twice over 12-24 hours in MEMalpha (see point 1.), conclusions about growth phase are not easy to draw.

Response: Based on the reviewer’s comment, we include the data by growing C. jejuni in MH media in Figure 3c and d. 

  • Line 159 (“data not shown”): data for Fe2+ should be added as a second group of bars in Fig. 1.

Response: We included the data for Fe2+ in the revised version (Figure 1b).

  • The authors should discuss their findings with respect to the concentration of iron in the intestinal environment.

Response: We discussed it in the revised version (Lines 277-291)

Minor points:

  • Line 103: The construction of the cosR overexpression strain was described in reference [13] rather than [17].

Response: The detailed method about the construction was provided in reference [13], where we constructed a CosR overexpression strain using a kanamycin resistance cassette. Because the selection of pMW10 marker is kanamycin, we had to construct another overexpression strain in reference [17] using a chloramphenicol cassette. The methods for constructing the two overexpression strains are the same except for the resistance marker. In this study, we used the overexpression strain carrying a chloramphenicol marker, which was used in reference [17], using the method described in reference [13]. That is why we cited the two references.

  • Lines 249-250: The authors should add some notion about the magnitude of iron affecting cosR and perR expression.

Response: We described it in the revised version (Lines201-204)

References

  1. Barnawi, H.; Masri, N.; Hussain, N.; Al-Lawati, B.; Mayasari, E.; Gulbicka, A.; Jervis, A.J.; Huang, M.H.; Cavet, J.S.; Linton, D. RNA-based thermoregulation of a Campylobacter jejuni zinc resistance determinant. PLoS Pathog 2020, 16, e1009008, doi:10.1371/journal.ppat.1009008.
  2. Oh, E.; Andrews, K.J.; Jeon, B. Enhanced biofilm formation by ferrous and ferric iron through oxidative stress in Campylobacter jejuni. Front Microbiol 2018, 9, doi:10.3389/fmicb.2018.01204.
  3. Wang, H.; Zeng, X.; Lin, J. Enterobactin-specific antibodies inhibit in vitro growth of different Gram-negative bacterial pathogens. Vaccine 2020, 38, 7764-7773, doi:https://doi.org/10.1016/j.vaccine.2020.10.040.
  4. Palyada, K.; Threadgill, D.; Stintzi, A. Iron acquisition and regulation in Campylobacter jejuni. J Bacteriol 2004, 186, 4714-4729, doi:10.1128/jb.186.14.4714-4729.2004.

Reviewer 2 Report

General: The article describes a very interesting and important subject. The manuscript describes the modulation and control of oxidative stress defence in Campylobacter, specifically at the level of regulation of perR transcription. The idea of this manuscript is very clear and covers one of the most important survival mechanisms of Campylobacter, which usually depends on their stress response.

The abstract does not contain enough clear information about the purpose and methods used, so it can be partially misleading even when mentioning gene expression.

Most of the deficiencies are found mainly in the Materials and Methods chapter. The authors have used many methods that are only mentioned and partially described in the results. I think that the methods need to be described clearly, maybe even briefly, in the materials section

Iron is very important, as the authors have shown. Perhaps a more detailed discussion of how iron relates to the stress response is missing - not just modulation, but a concrete relationship with oxidative stress response and further Campylobacter survival/virulence.

 Why was Minimum Essential Medium used for the cultivation of Campylobacter.

How was the concentration of iron in the medium regulated? There is nothing written about this in the materials.

What methodology was used to track the growth curve - this should be described in more detail in the materials. 

Author Response

  • The abstract does not contain enough clear information about the purpose and methods used, so it can be partially misleading even when mentioning gene expression.

Response: We modified the abstract in the revised version.

  • Most of the deficiencies are found mainly in the Materials and Methods chapter. The authors have used many methods that are only mentioned and partially described in the results. I think that the methods need to be described clearly, maybe even briefly, in the materials section

Response: According to the comment, we modified the materials and methods section.

  • Iron is very important, as the authors have shown. Perhaps a more detailed discussion of how iron relates to the stress response is missing - not just modulation, but a concrete relationship with oxidative stress response and further Campylobacter survival/virulence.

Response: We discussed it in the revised version (Lines 277-291)

  • Why was Minimum Essential Medium used for the cultivation of Campylobacter.

Response: We explained why MEM was used in the experiment with references (Lines 179-182). We cited only a few representative studies done by different research groups.

  • How was the concentration of iron in the medium regulated? There is nothing written about this in the materials.

Response: We described it in the revised version (Lines 101-103)

  • What methodology was used to track the growth curve - this should be described in more detail in the materials. 

Response: Our laboratory routinely cultures C. jejuni using the same growth conditions, and the growth phases are well characterized under the conditions used in the study, where C. jejuni enters the exponential phase after 3 hours and the stationary phase after 12 hours. We have used this culture method for more than 10 years. In the revised version, we included the data using MH media in Figure 3 and had to remove the growth curve because of the space limit.

Round 2

Reviewer 1 Report

The authors addressed all points, except in figure 3, the growth curves are missing. Please add as in the previous version and include the growth of C. jejuni in MH. In the legend of Fig. 3, those OD values are still mentioned.

Author Response

The authors addressed all points, except in figure 3, the growth curves are missing. Please add as in the previous version and include the growth of C. jejuni in MH. In the legend of Fig. 3, those OD values are still mentioned.

Response: According to the reviewer’s comment, we included the growth curves in the revised version. Thank you.